# Study on Mechanical Properties of AlFeCrMoNi$_{1.8}$Nb$_{1.5}$ Eutectic High-Entropy Alloy Coating Prepared by Wide-Band Laser Cladding

**Feng Li** [1], **Shisong Zheng** [2] **and Fang Zhou** [2,*]

1   College of Materials and Metallurgy, Guizhou University, Guiyang 550025, China; l15335520187@gmail.com
2   China Zhenhua Group Xinyun Electronic Company & Development Co., Ltd. (State-run Factory 4326), Guiyang 550018, China; zss5661@163.com
*   Correspondence: zss7451@163.com

**Abstract:** In this study, AlFeCrMoNi$_{1.8}$Nb$_{1.5}$ (at.%) eutectic high-entropy alloy (EHEA) coating was successfully prepared on the surface of M2 high-speed steel (HSS) by wide-laser cladding. The effects of laser defocusing amount, laser power, scanning speed, and preset powder thickness on the formation quality of the EHEA coating were studied by the orthogonal experimental design, then the mechanical properties of the coating prepared by water-cooled solidification under optimal process parameters were studied. The experimental results showed that the optimal laser cladding process parameters are defocusing with an amount of −30 mm, laser power of 4 kW, scanning speed of 3 mm/s, and preset powder thickness of 1.5 mm. The substrate exhibited a favorable metallurgical bond with the coating, characterized by a stable interface devoid of any holes or cracks. Furthermore, the coating, which was prepared using water cooling, displayed a finer lamellar eutectic structure comprising FCC and Laves phases. The microhardness of the coating (544 HV$_{0.2}$) was significantly higher than that of the substrate M2 HSS (~220 HV$_{0.2}$), accompanied by good wear resistance.

**Keywords:** eutectic high-entropy alloy; mechanical properties; strengthening mechanisms; wide-band laser cladding

## 1. Introduction

In the past few decades, high-speed steel has been widely used in the modern manufacturing industry as the production material of cutting tools due to its relatively low price and excellent thermal hardness at 600 °C. However, high-speed steel could only work in low-speed cutting [1] and the shortcomings of poor toughness limited its application. Therefore, it is necessary to find ways to improve the performance of M2 HSS. Surface modification of cutting tools was a more effective and economical means to improve overall performance, such as thermal diffusion, boron-doping, physical vapor deposition (PVD), and chemical vapor deposition (CVD) [2–9]. However, the coatings prepared by these methods had the disadvantages of poor toughness, limited thickness, and low bonding strength with the substrate, which limited their application at ultra-highspeed cutting [10]. Therefore, selecting the appropriate surface modification technology is necessary to enhance the matrix properties and bonding strength.

The coating prepared by laser cladding technology had high bonding strength with the substrate, controllable thickness, uniform microstructure, and high hardness [11,12], making it stand out in many current material surface modification methods. It is worth mentioning that the laser cladding process parameters have a great influence on the formability and performance of the coating. Zhang et al. [13] studied the influence of laser cladding process parameters on the forming size of IN718 cladding alloy by orthogonal experimental design, and analyzed the interface structure and mechanical properties of the cladding layer under the optimum process parameter. The results showed that the

optimum laser cladding process of IN718 alloy coating is as follows: scanning speed was 6 mm/s, laser power was 900 W, powder feeding amount was 10 g/min, and the average interfacial shear strength of laser cladding IN718 alloy coating was 608.87 MPa. Gong [14] studied the effects of laser power, scanning speed, spot diameter, and powder feeding rate on the width and height of single-channel cladding by a four-factor three-level orthogonal experiment. The results showed that the influence of process parameters on the cladding width showed the following order: laser power (2.5 kW) > scanning speed (10 m/s) > powder feeding rate (21 g/min) > spot diameter (5 mm), and the influence on cladding height showed the following order: scanning speed (12 m/s) > laser power (3 kW) > powder feeding rate (21 g/min) > spot diameter (4 mm). By investigating the effect of defocusing amount on the quality of the Fe-based powder cladding layer, Gao et al. [15] found that a coating with less porosity and lower dilution rate can be obtained under a negative defocusing amount (the spot is below the focal spot location). Because the laser energy distribution was more uniform under negative defocus, on the contrary, the laser energy distribution was annular under positive defocus (The spot is above the focal spot position). However, many researchers [13–15] have only studied cladding coatings under circular laser spots. Compared with the circular laser spot, the rectangular laser spot had a more uniform energy distribution, which reduced the dilution rate during laser cladding [16] and is often overlooked. In fact, wide-band laser showed significant advantages in laser cladding [17–19]. Yao et al. [20] studied the preparation of WC-NiCrMo alloy powder coating on SS316 surface by different laser melting techniques. The results showed that the wide-band laser cladding coating had more uniform energy distribution and laser power density than the circular laser spot, so as to obtain better microstructure uniformity and better wear resistance.

HEAs are a new type of metal material proposed by Yeh et al. [21,22] and Cantor et al. [23] in 2004. It is composed of four or more alloying elements with an atomic percentage of 5~35 at.%. HEAs have four effects: high-entropy effect, lattice distortion effect, cocktail effect, and sluggish diffusion effect. Therefore, it has high wear resistance [24–26], excellent thermal stability [27–29], high-temperature oxidation resistance [30,31]. HEAs usually tend to form simple solid solution structures, such as soft face-centered cubic (FCC) [32], hard body-centered cubic (BCC) [33], and hexagonal close-packed (HCP) [34]. Because the single-phase structure of HEA is difficult to meet the actual needs, EHEAs composed of two or more phases to increase the phase boundary and effectively reduce the dislocation movement have become a hot research point in recent years. For example, based on the eutectic structure of the $CoFeNi_2V_{0.5}Nb_{0.75}$ HEA [35] with the FCC + Laves ($Fe_2Nb$) phases, after quenching at 800 °C, it exhibited the best comprehensive mechanical properties of fracture strength (2586 MPa), yield strength (2075 MPa), and plastic strain (16.7%), respectively. $Co_{25.1}Cr_{18.8}Fe_{23.3}Ni_{22.6}Ta_{8.5}Al_{1.7}$ EHEA composed of FCC + C14 Laves phases was prepared by the powder metallurgy method by Han et al. [36]. The nano-L12 phase (4~5 nm) in the FCC matrix plays a strengthening role, and the equiaxed Laves phase also helps to alleviate stress concentration and fracture anisotropy, the microstructure of the alloy is highly stable and still has good high-temperature strength after annealing at 1000 °C for 100 h.

Based on the findings from Kuang [37], it was observed that Al elements tend to vaporize during the laser cladding process, leading to the enrichment of Al on the coating surface and the formation of an uneven structure. This gasification of Al occurs due to the presence of high melting point components in the composition, which require greater energy for laser cladding. To mitigate Al vaporization, we eliminated high melting point components such as Ti and W and substituted Al with commercially available nickel-coated aluminum composite powder (Ni:Al = 1.83:1; at.%), thereby reducing the vaporization of Al. The addition of Ni also helps to minimize crack formation [38] and promotes a more uniform microstructure. Furthermore, the combination of Ni and Al readily forms a hard B2-NiAl phase within the coating, exhibiting excellent resistance to softening during high-temperature annealing experiments. In this study, we investigated the process parameters

for the improved AlFeCrMoNi$_{1.8}$Nb$_{1.5}$ coating, aiming to obtain EHEA coatings with satisfactory macroscopic quality and superior mechanical properties while addressing the challenges associated with Al vaporization and uneven microstructure. The considered laser cladding process parameters included defocusing amount, laser power, scanning speed, and preset powder thickness.

## 2. Materials and Methods

### 2.1. Materials and Sample Preparation

The alloy of nominal chemical composition AlFeCrMoNi$_{1.8}$Nb$_{1.5}$ (denoted as: Ni$_{1.8}$Nb$_{1.5}$) was prepared by wide-band laser cladding. Raw elemental powders (>99.5% pure; Changsha Tijo Metal Material Co., Ltd., Changsha, China; Specifications of −150~300 mesh, Ni and Al powder replaced by Ni-coated Al composite powder; composition demonstrated in Table 1) were placed in a steel grinding tank containing some steel balls with different diameters of 6–15 mm, and then mixed at 300 r/min for 0.5 h in a QM-3SP2 ball mill. The commercial M2 HSS (chemical formula: W$_6$Mo$_5$Cr$_4$V$_2$; Heye Special Steel Co Ltd., Shijiazhuang, China; composition demonstrated in Table 1) with the dimensions of φ 50 × 10 mm was selected as the substrate for the coating, and the rust and grease on the surface of M2 HSS were polished and cleaned with a grinding machine for standby. The powder mixture was pre-placed on the M2 HSS surface to prepare the Ni$_{1.8}$Nb$_{1.5}$ coating using a YLS-6000-type fiber laser (Wuhan Raycus Fiber Laser Technologies Co., Ltd., Wuhan, China) with a rectangular laser spot size of 10 × 2 mm, a focal length of 30 mm, and argon shielding gas with a flow rate of 25 L/min. The wide-band laser cladding diagram is shown in Figure 1a. Specimens were fabricated into different sizes by wire electrical discharge machining, then ground by SiC abrasive paper (100~2000 mesh) and polished with diamond polishing agents (W 2.5 μm) for microstructure analysis and mechanical properties testing. The metallographic samples were etched by HNO$_3$:HCl = 1:3 mixed solution.

**Table 1.** The chemical compositions of M2 and Ni$_{1.8}$Nb$_{1.5}$.

| W$_6$Mo$_5$Cr$_4$V$_2$ | | AlFeCrMoNi$_{1.8}$Nb$_{1.5}$ | | |
|---|---|---|---|---|
| Element | wt.% | Element | wt.% | at.% |
| W | 5.50~6.75 | Al | 0.12 | 0.38 |
| Mo | 4.50~5.50 | Fe | 11.73 | 18.11 |
| Cr | 3.80~4.40 | Cr | 10.93 | 18.11 |
| V | 1.75~2.20 | Mo | 20.16 | 18.11 |
| C | 1.05~1.20 | Nb | 29.29 | 27.16 |
| Al | 0.80~1.20 | | | |
| Si | ≤0.60 | Ni-coated Al | | |
| Mn | ≤0.40 | composite | 27.75 | 18.11 |
| S | ≤0.03 | powder | | |
| P | ≤0.03 | | | |

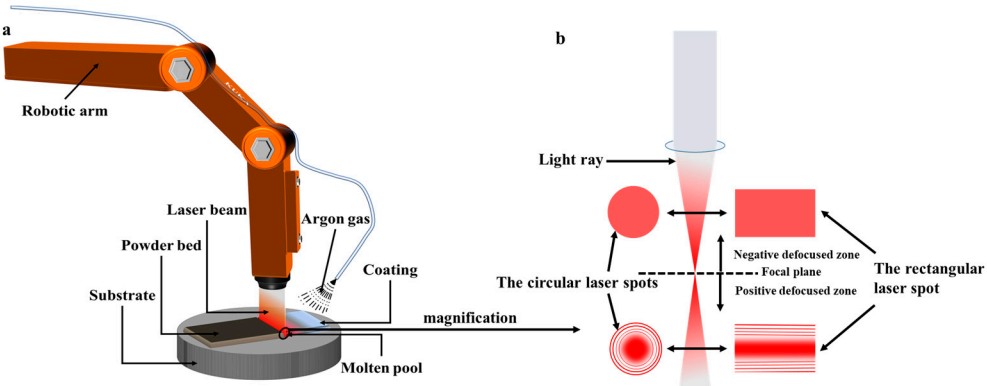

**Figure 1.** (**a**) The laser cladding diagram; (**b**) Enlarged view of working plane.

### 2.2. Laser Cladding Process Parameters Design

The orthogonal experiment is an efficient and rapid test method to study multifactor and multi-level parameters. In order to study the influence of the laser cladding process parameters on the forming quality and mechanical properties of the $Ni_{1.8}Nb_{1.5}$ alloy coating, and to prepare the alloy coating with good bonding with the substrate and no defect, we used the orthogonal test method. Four factors: defocus amount (mm), laser power (kW), scanning speed (v mm/s), and preset powder thickness (T mm), were selected. Three levels were taken for each factor. The parameters are shown in Table 2. Nine groups of laser cladding experiments with different process parameters were designed.

**Table 2.** The factors and levels of orthogonal experiments.

| Number | Defocus Amount (mm) | Laser Power (KW) | Scanning Speed (mm/s) | Preset Powder Thickness (mm) |
|--------|--------------------|-----------------|----------------------|------------------------------|
| 1 | +30 | 3.5 | 3 | 1 |
| 2 | 0 | 4 | 4 | 1.2 |
| 3 | −30 | 4.5 | 5 | 1.5 |

Notes: +30, 0, and −30, respectively, represent the working planes under positive defocused zone distances from the focal plane of 3 mm, and 0 mm, and the working planes under negative defocused zone distances from the focal plane of 3 mm.

### 2.3. Microstructure Characterization

Phase identification was performed using Bruker AXS D8 Advance X-ray diffraction (XRD, Wuhan Shuosibai Testing Technology Co., Ltd. Wuhan, China) equipped with Cu Ka radiation at a scanning speed of $10°/min$ with a voltage of 40 kV and current of 40 mA. The microstructure observations and elemental analysis were studied using ZEISS Gemini 300 scanning electron microscope (SEM, Carl Zeiss AG, Jena, Germany) with an energy dispersive spectroscopy (EDS) setup.

### 2.4. Microhardness and Wear Performance Experiments

The Micro-Vickers hardness tester (JMHV-1000, Hong Kong, China) was used to measure the microhardness. The test parameters were load 2N ($HV_{0.2}$), and the loading time 15 s. The friction properties of the coating were tested at room temperature using an MFT-5000 multi-functional friction and wear tester. The normal load was 50 N, the length of the wear track was 4 mm, the sliding speed was 400 r/min, the test time per sample was 15 min, and the friction pair material was SiC with a diameter of 6 mm. The 3D morphology of wear tracks was characterized by Contour GT-K three-dimensional optical profiler (BRUKER, Billerica, MA, USA).

$$\Delta V = S \times l \tag{1}$$

Using Equation (1) to Calculate Wear Volume.

## 3. Results

### 3.1. Macroscopic Morphology of Coatings

The macroscopic morphology of the surface of the nine groups of laser cladding coatings is depicted in Figure 2. Each row represents the macroscopic morphology of the coatings prepared under the same defocusing amount, with laser power gradually increasing, different scanning speeds, and powder thickness. Each column corresponds to the macroscopic morphology of the coatings prepared under the same laser power, varying defocusing amount from positive to negative, different scanning speeds, and preset powder thickness. Upon comparing the samples within each row, it is observed that when the defocusing amount remains constant, there is no significant change in the macroscopic morphology of the coatings with increasing laser power. Comparing each column, it can be

concluded that as the defocusing amount shifts from positive to negative, the macroscopic surface morphology of the cladding coatings tends to become smoother and flatter.

Under positive defocusing conditions, the energy density from the center of the beam to the edge decreases, resulting in insufficient melting of the powder at the edge of the cladding track. As a result, the unmelted portion forms a non-metallic bond with the surface area of the M2 HSS substrate, leading to the appearance of holes or defects during the coating shrinking process with the cold substrate [15]. The interaction between the high-energy laser beam and metal powder involves a vigorous thermal process. In the case of positive defocusing, the laser spot becomes larger, causing an expansion of the molten pool and increased droplet splashing. Consequently, the surface of the cladding coatings appears uneven when using a positive defocusing amount. Conversely, under negative defocusing conditions, the laser beam energy is uniformly concentrated, resulting in a dense and smooth surface of the prepared coating.

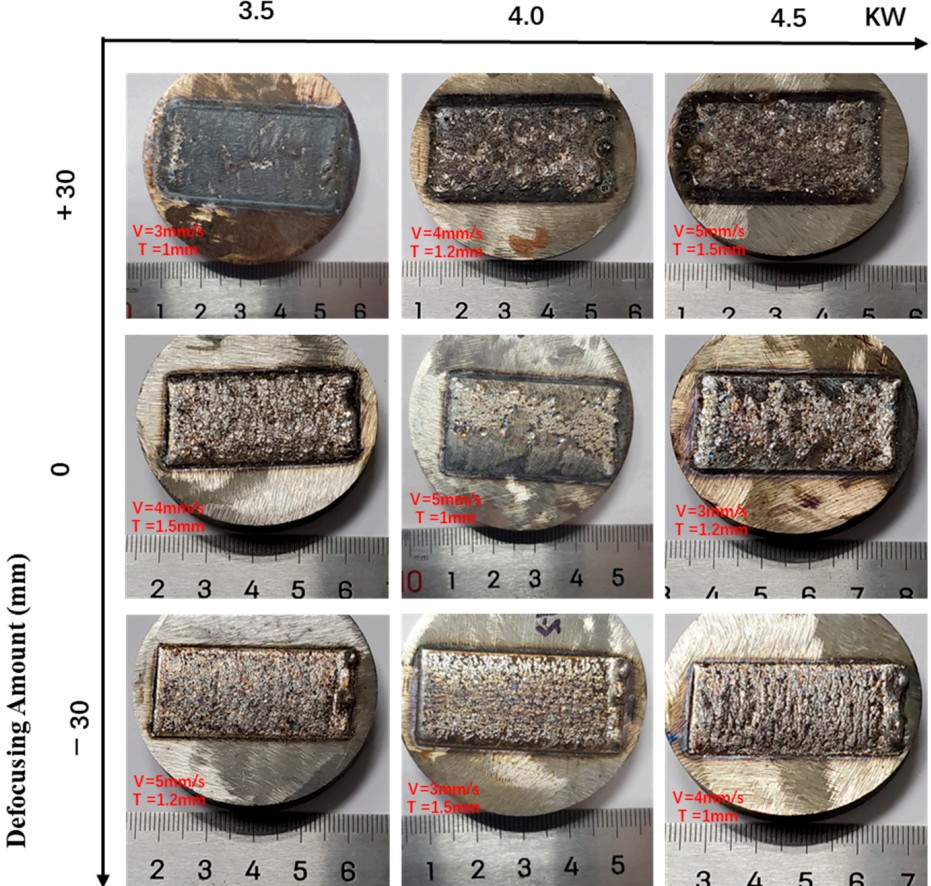

**Figure 2.** The macroscopic morphology of the surface of the 9 groups of laser cladding coatings.

Figure 3 illustrates the cross-sectional morphology of the coatings prepared using different process parameters. When the defocusing amount is reduced, the coating thickness increases while maintaining the same preset powder thickness. This can be attributed to the concentration of laser beam energy after reducing the defocusing amount, allowing the formed molten pool to more effectively capture metal powder. Consequently, this reduces splattering and facilitates the formation of a more uniform, dense, and thicker coating. Notably, coatings exhibit the presence of pores when a positive defocus is employed. Moreover, when the laser power is increased to 4.5 kW, the coating becomes susceptible to cracking. This phenomenon occurs due to the enlargement of the molten pool size associated with the higher power. Consequently, the thermal stress generated during the solidification process intensifies with the increased molten pool area. In cases of rapid cooling, the thermal stress cannot be relieved within the liquid molten pool and

subsequently be released after solidification. This, in turn, generates tensile stress on the coating, providing the driving force for crack formation.

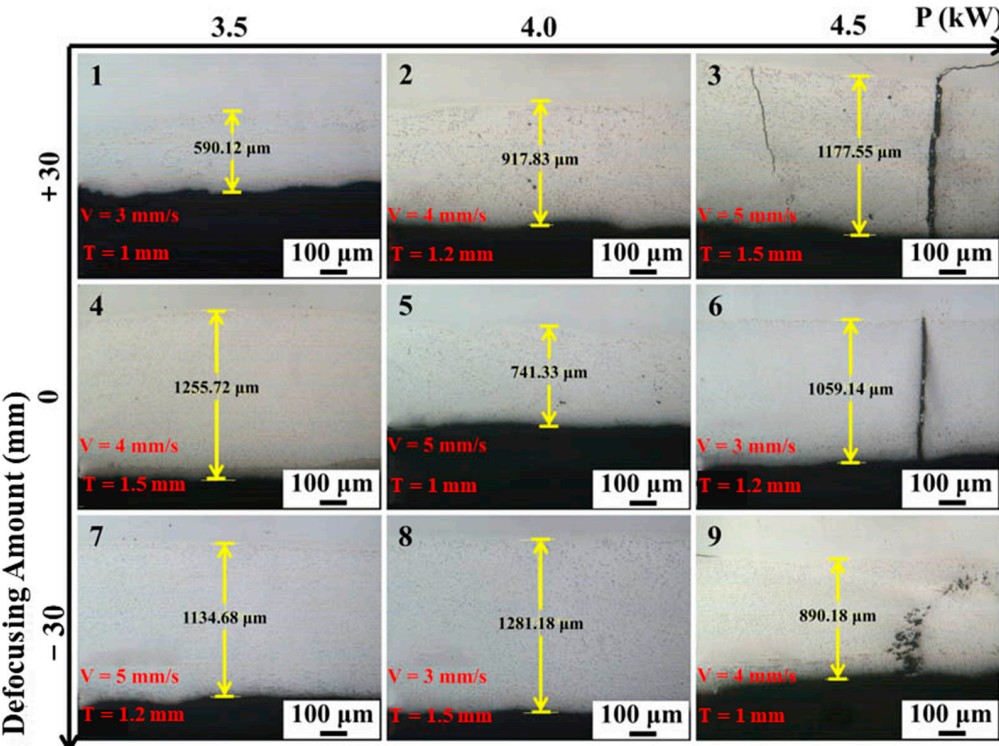

**Figure 3.** The cross-sectional morphology of the coatings.

The results of the orthogonal experiments are presented in Table 3. From these results, it can be intuitively concluded that the macroscopic morphology of the coatings is primarily influenced by the defocusing amount. It is observed that obtaining a coating with a smooth surface is easier when using negative defocusing. Positive defocusing leads to an uneven surface and the occurrence of defects such as insufficient melting, which results in holes in the coating. Another factor that significantly affects the coatings is the powder thickness, which mainly determines the thickness of the coatings. Additionally, the laser power and scanning speed collectively determine the energy input. It is important to note that higher laser power can increase the thermal stress of the molten pool, which may cause cracks in the coatings.

**Table 3.** The results of orthogonal experiments.

| Number | Defocus Amount (mm) | Laser Power (kW) | Scanning Speed (mm/s) | Preset Powder Thickness (mm) | Macroscopic Morphology | Formability |
|---|---|---|---|---|---|---|
| No. 1 | +30 | 3.5 | 3 | 1 | Uneven | Holes-rich and thinner coating |
| No. 2 | +30 | 4 | 4 | 1.2 | Uneven | Holes |
| No. 3 | +30 | 4.5 | 5 | 1.5 | Uneven | Cracks and holes |
| No. 4 | 0 | 3.5 | 4 | 1.5 | Uneven | Good combination and no defects |
| No. 5 | 0 | 4 | 5 | 1 | Uneven | Thinner coating |
| No. 6 | 0 | 4.5 | 3 | 1.2 | Uneven | Cracks |
| No. 7 | −30 | 3.5 | 5 | 1.2 | Slightly even | Good combination and no defects |
| No. 8 | −30 | 4 | 3 | 1.5 | Even and smooth | Good combination and no defects |
| No. 9 | −30 | 4.5 | 4 | 1 | Slightly even | Holes-rich and thinner coating |

Based on the optimization of process parameters, a coating with desirable characteristics, including a smooth surface, absence of holes and cracks, and a thickness of 1281.18 μm, was successfully prepared. The optimized process parameters for laser cladding were determined as follows: laser power of 4 kW, defocusing amount of −30 mm, scanning speed of 3 mm/s, and powder thickness of 1.5 mm. Therefore, it is determined as the optimizing laser cladding process parameters, and subsequent experiments are completed under the process.

### 3.2. Effect of Cooling Rate on Coating

On the basis of optimizing the process parameters, we further studied the effect of the cooling rate on the microstructure of the alloy. The coating of water-cooled solidification is to put the substrate into the water at room temperature to make the substrate surface 2~3 mm above the horizontal plane for laser cladding. Figure 4 is the microstructure morphology of the coatings prepared by air cooling and water cooling at room temperature under the optimal laser cladding process parameters. Figure 4a is the air-cooled coating microstructure. It can be seen that the middle part of the coating is a coarse dendrite structure, and the lamellar eutectic structure is near the substrate and the top of the coating. This is because the heat dissipation in the middle of the coating is slow and the nucleation rate is low so that the grains grow and form dendrites. The heat dissipation near the substrate and the surface of the coating is faster than that in the middle of the coating, forming a small amount of fine eutectic structure. Figure 4b depicts the microstructure of the water-cooled coating, revealing a uniformly finer lamellar eutectic structure throughout the entire coating. This outcome is due to the increased cooling rate of the alloy, which enhances undercooling. Improved undercooling promotes nucleation and leads to grain refinement. At the same time, it also promotes the occurrence of pseudo-eutectic transformation and obtains more eutectic structures.

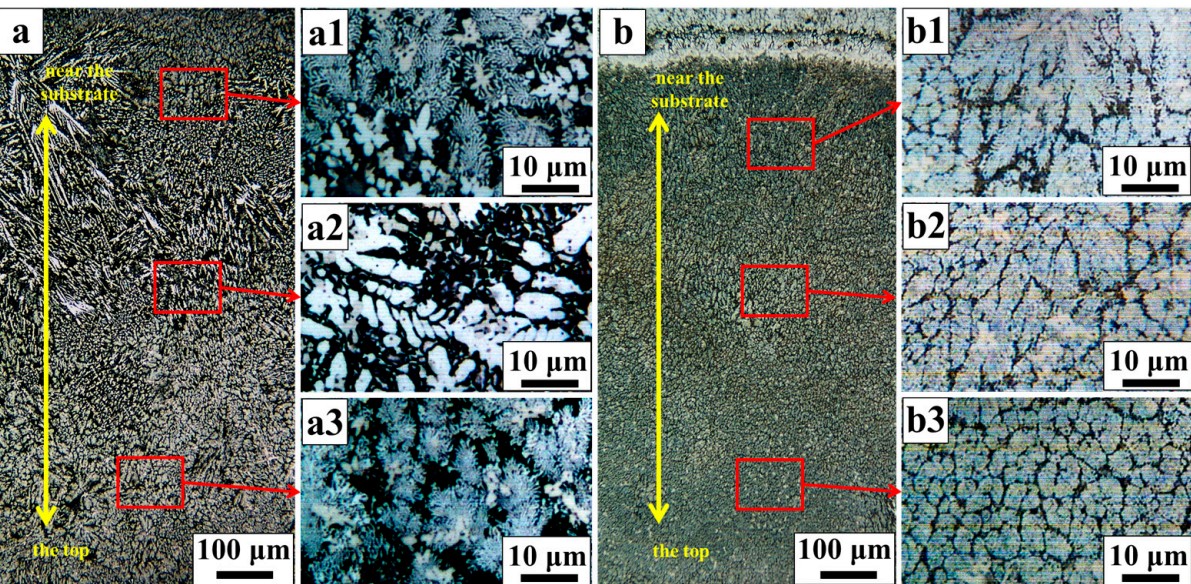

**Figure 4.** Effect of cooling rate on the microstructure of $Ni_{1.8}Nb_{1.5}$ alloy coatings: (**a**) air cooling (**a1**–**a3**) for the partial enlargement in (**a**); (**b**) water cooling (**b1**–**b3**) for the partial enlargement in (**b**).

Figure 5 shows the microhardness curves of the coatings prepared by air cooling and water cooling during the optimal laser cladding process. The testing rule is from the coating surface to the base material area, and multiple measurements are averaged for reflecting the true microhardness of the coatings. The average microhardness of the air-cooled (537 $HV_{0.2}$) is lower than that of the water-cooled (544 $HV_{0.2}$). The weakening of the lamellar boundary strengthening effect of the eutectic structure in the bonding zone

(BZ; In the laser cladding process, the bottom temperature of the coating is higher, and the grains are easy to grow. The G/R ratio is small, resulting in a small zone where the grains grow in the form of planar crystals called the bonding zone, and this is due to the fact that the high-temperature melting makes the base material elements penetrate into the bottom of the coating, causing the composition in the bonding zone to deviate from the eutectic composition of the coating, resulting in a decrease in hardness. The reason for the increase in the hardness of the heat-affected zone (HAZ; the zone near the surface of the substrate where the hardness changes when laser cladding) is that the substrate undergoes a martensitic transformation to achieve high microhardness (~850 $HV_{0.2}$ and ~750 $HV_{0.2}$). The substrate′s microhardness, unaffected by laser processing, is approximately 220 $HV_{0.2}$. Overall, the water-cooled solidification method enhances the microhardness of the laser-cladding coating.

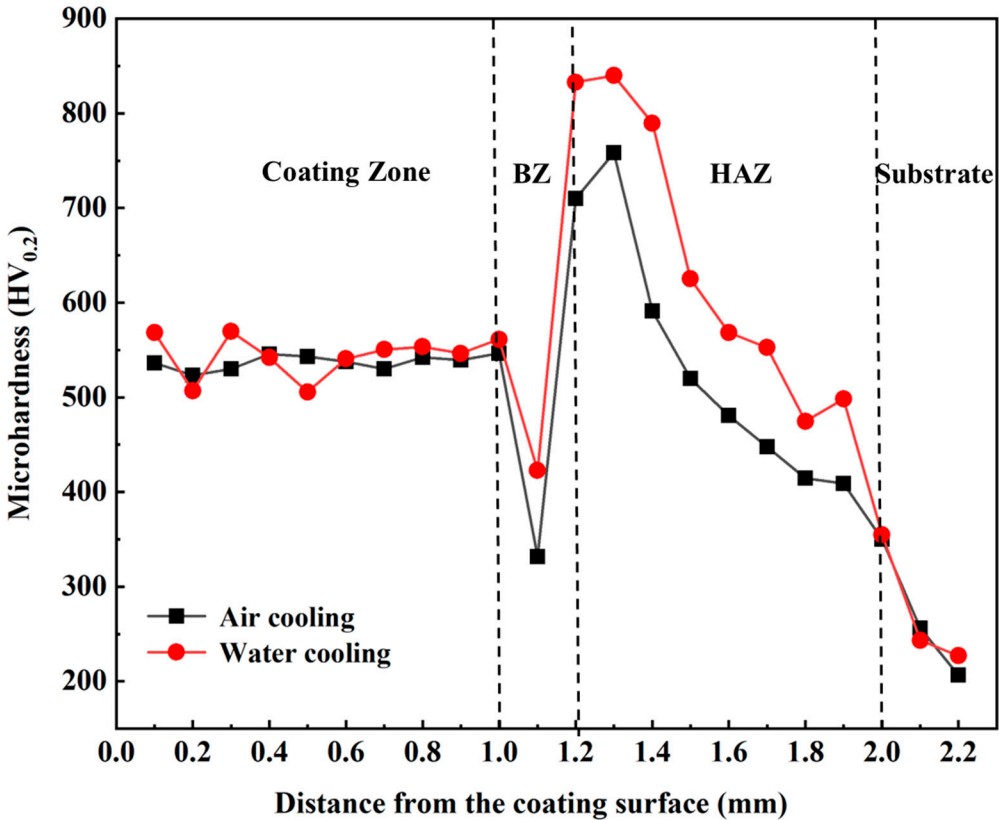

**Figure 5.** The microhardness of water cooling and air cooling.

　　The cross-sectional characterization of the $Ni_{1.8}Nb_{1.5}$ EHEA coating, prepared using optimized laser cladding process parameters and water cooling, is depicted in Figure 6. The cross-sectional SEM and elemental content distribution of the $Ni_{1.8}Nb_{1.5}$ EHEA coating are shown in Figure 6a. It can be seen that even if the solidification process of the alloy is accelerated by water cooling, the coating still has no cracks and obvious holes. The coating prepared using Ni and Al powder tend to vaporize, resulting in Al segregation on the coating surface due to the high temperatures generated during the laser cladding process [37], whereas the coating with a more uniform composition can be obtained by using Ni-coated Al composite powder. A bonding band is formed between the substrate and the coating [39], indicating a strong metallurgical bonding between the substrate and the coating under the preparation method, as shown in Figure 6b.

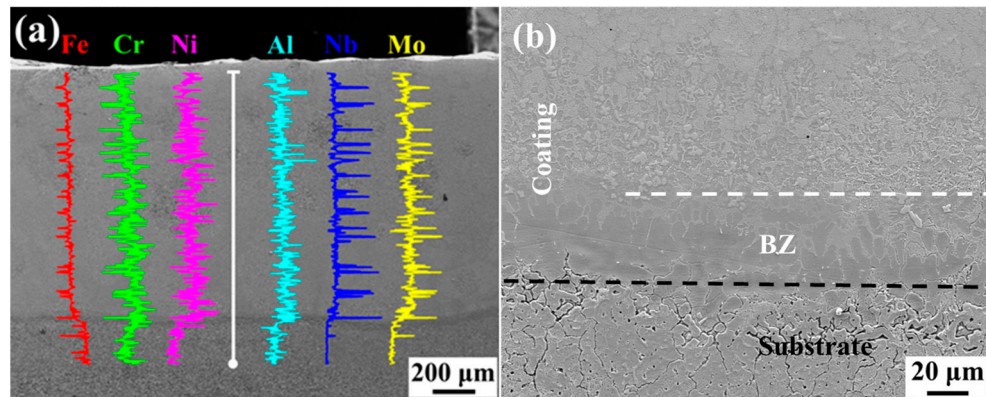

**Figure 6.** (**a**) The cross-sectional SEM images and elemental analysis of the Ni$_{1.8}$Nb$_{1.5}$ EHEA coating; (**b**) The bonding zone.

### 3.3. Phase and Microstructure Analysis

Figure 7 illustrates the phase composition, microstructure, and micro-regional component analysis of the Ni$_{1.8}$Nb$_{1.5}$ EHEA coating, which was prepared using optimized cladding process parameters with water-cooled solidification. In Figure 7a, the XRD pattern of the Ni$_{1.8}$Nb$_{1.5}$ EHEA coating is presented, consisting of several phases: FCC matrix phase-(Fe, Ni) (PDF #47-1417), Laves phase-Fe$_2$Nb (PDF #17-0908), B2 phase-NiAl (PDF # 44-1188), and Nb-rich carbides (PDF #38-1364).

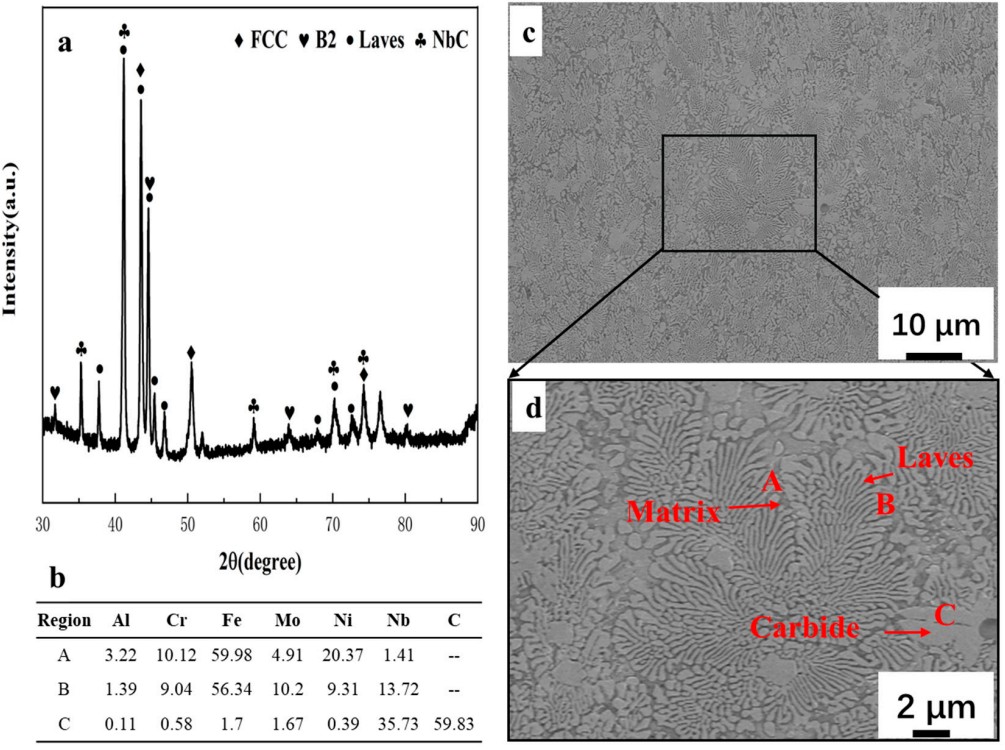

| Region | Al | Cr | Fe | Mo | Ni | Nb | C |
|--------|------|-------|-------|------|-------|-------|-------|
| A | 3.22 | 10.12 | 59.98 | 4.91 | 20.37 | 1.41 | -- |
| B | 1.39 | 9.04 | 56.34 | 10.2 | 9.31 | 13.72 | -- |
| C | 0.11 | 0.58 | 1.7 | 1.67 | 0.39 | 35.73 | 59.83 |

**Figure 7.** Phase microstructure and micro-regional component analysis of the Ni$_{1.8}$Nb$_{1.5}$ EHEA coating; (**a**) XRD pattern of the Ni$_{1.8}$Nb$_{1.5}$ EHEA coating; (**b**) micro-regional component content (at.%) of (**d**); (**c**) The SEM image of the Ni$_{1.8}$Nb$_{1.5}$ EHEA coating; (**d**) enlarged view of the rectangular region in (**c**).

The SEM diagram of the coating, shown in Figure 7c, reveals a fine and dense eutectic layer structure distributed throughout the coating, with randomly embedded massive gray particles. Analyzing Figure 7b,d, it becomes apparent that the gray elongated structure (region B in Figure 7d) is abundant in Fe and Nb, both of which exhibit a relatively negative mixing enthalpy ($\Delta Hmix_{Fe2Nb}$ = −20.85 kJ/mol). Combining this observation with Figure 7a, we can conclude that this structure corresponds to an intermetallic compound rich in $Fe_2Nb$. The intergranular structure (region A in Figure 7d) is rich in Fe and Ni, indicating the presence of the FCC matrix phase in conjunction with Figure 7a. The eutectic structure comprises alternating strips of the FCC matrix phase and the Laves phase. Finally, based on Figure 7d, the massive gray particles (region C) are identified as NbC particles, given their richness in Nb and C elements, as confirmed by Figure 7a.

### 3.4. Mechanical Properties and Characterizations

Figure 8 represents a summary graph that illustrates the mechanical properties of the $Ni_{1.8}Ni_{1.5}$ EHEA coating and M2 HSS at room temperature. In Figure 8a, the coating is presented with an average microhardness value of 544 $HV_{0.2}$, which is significantly higher than that of M2 HSS (~220 $HV_{0.2}$). The outstanding performance can be attributed to a synergistic effect resulting from multiple factors. First, the fine grains formed under the rapid heating and rapid cooling process of the coating during laser cladding (as depicted in Figure 4b) contribute to fine grain strengthening, and increase in interface density which prevent dislocation motion. Additionally, the formation of a dense and fine lamellar eutectic structure strengthens the lamellar boundaries. Furthermore, hard particles like NbC and NiAl, which are uniformly embedded in the matrix, enhance diffusion strengthening. Figure 8b,c display the sliding friction coefficient curves and sliding wear volume loss histogram of the coating and the substrate M2 HSS at room temperature. The data clearly show that the friction coefficient of the coating is significantly lower than that of M2 HSS, and the sliding wear volume loss of the coating is also lower. This result is directly related to the average microhardness of the two materials, which aligns with the Archard law [40]. Section morphology curves and the 3D morphology of the wear track images are shown in Figure 8d–f. The wear track depth of the EHEA coating is lower and relatively flat compared to that of M2 HSS. Additionally, the shape of the wear curve for M2 HSS tends to be an arc in Figure 8d. This can be effectively explained by the presence of a fine lamellar eutectic structure, hard phases, and finer grains in the coating during the wear process, which enhances its ability to resist the cutting of micro-protrusions on the material surface. Figure 8c shows that the wear volume loss of M2 HSS ($1.37 \times 10^{-1}$ $mm^3$) is obviously larger than that of the coating ($0.52 \times 10^{-1}$ $mm^3$). The soft M2 HSS exhibits weak resistance to shear stress during the friction process, leading to obvious parallel scratches in the gully, indicating that the surface wear mechanism of M2 HSS is primarily abrasive wear. Conversely, the 3D morphology of the EHEA coating is relatively flat, with small pits of varying depths in the gully, indicating that the harder coating has a stronger ability to resist wear. Therefore, it can be concluded that the wear mechanism of the coating is mainly adhesive wear. Overall, the mechanical properties of the coating prepared by laser cladding are superior to those of the substrate M2 HSS.

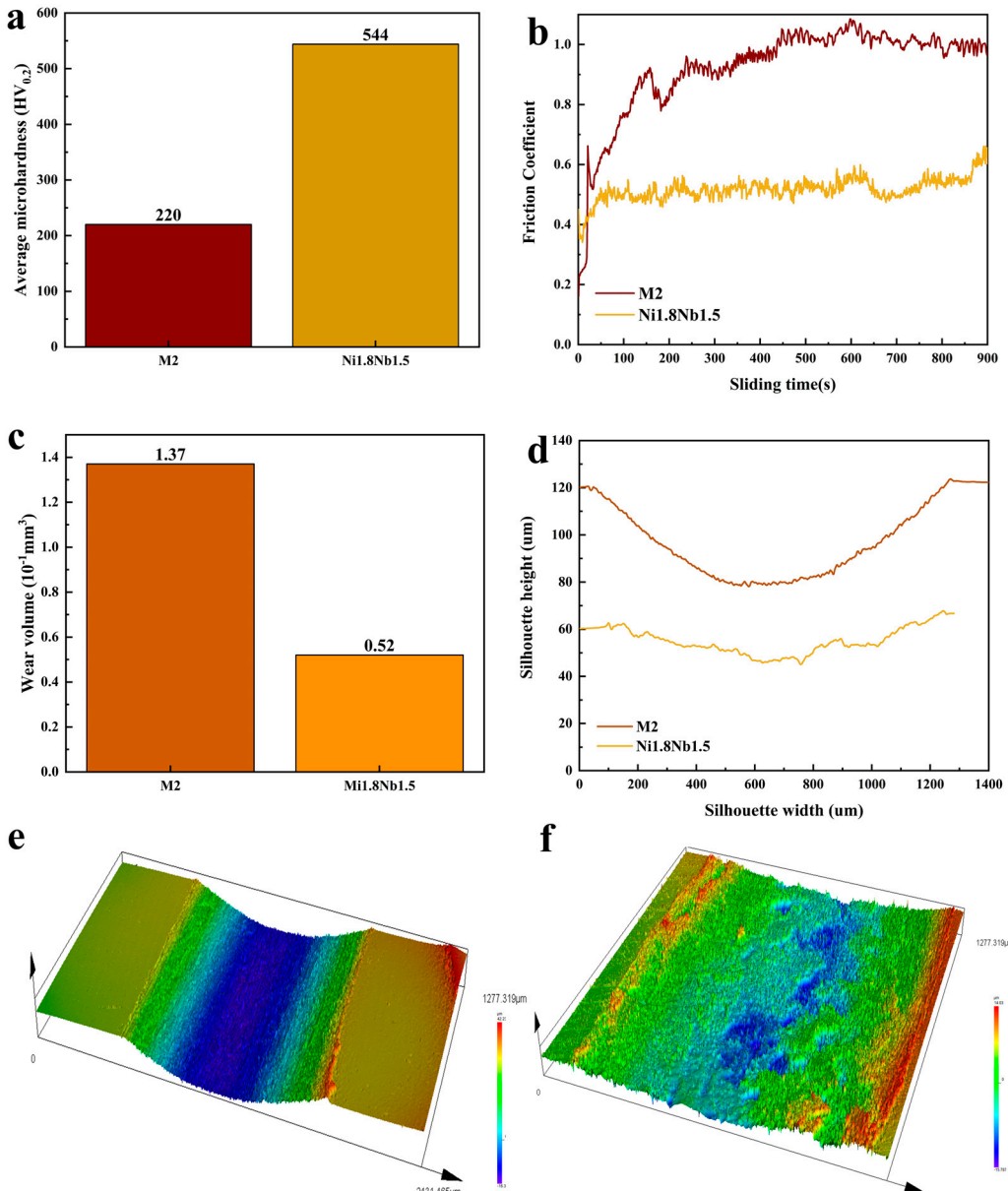

**Figure 8.** (**a**) average microhardness of M2 and the $Ni_{1.8}Nb_{1.5}$ coatings; (**b**) The Friction coefficient curves; (**c**) Wear volume loss histogram; (**d**) Section morphology curves of wear traces (M2 HSS and the $Ni_{1.8}Nb_{1.5}$ coating); (**e**,**f**) The 3D morphology of the wear tracks (M2 HSS and the $Ni_{1.8}Nb_{1.5}$ coating).

## 4. Discussion

### 4.1. HEA Criterion

Through the theoretical calculation of parameters such as mixing entropy ($\Delta Smix$), mixing enthalpy ($\Delta Hmix$), synergy parameter ($\Omega$), and atomic size difference ($\delta$) in HEAs, the approximate phase prediction of HEAs can be realized. The calculation formulas and the results of thermodynamic parameters are as follows in Table 4, where R is the universal gas constant (8.314 J/(mol·k)), $C_i$, $C_j$ is the atomic percentage content of the $i$th and $j$th elements, $\Delta H_{ij}^{mix}$ is the mixing enthalpy of the binary alloy, $r_i$ is the atomic radius of the $i$th element, and $\bar{r} = \sum_{i=1}^{n} C_i r_i$ is the average atomic radius, $T_m = \sum_{i=1}^{n} C_i (T_m)_i$, $(T_m)_i$ is the melting point of the $i$th component.

The thermodynamic calculation of the alloy satisfies the principle of forming a stable solid solution of multi-principal element HEA proposed by Professor Zhang [41]: $12 \leq \Delta Smix \leq 17.5$, $-22 \leq \Delta Hmix \leq 5$, $\Omega > 1.1$, $\delta \leq 6.6\%$, and the alloy system satisfies the definition of HEA ($\Delta Sconf > 1.61R$) [42].

**Table 4.** Calculation formulas [41,43,44] and results of thermodynamic parameters.

| Parameters | Computational Formulas | $Ni_{1.8}Nb_{1.5}$ |
|---|---|---|
| $\delta$ (%) | $\delta = 100\sqrt{\sum_{i=1}^{n} C_i \left(1 - \frac{r_i}{\bar{r}}\right)^2}$ | 6.44 |
| $\Delta Smix$ (J/(mol·K)) | $\Delta S_{mix} = -R \sum_{i=1}^{n} C_i \ln C_i$ | 1.7596R |
| $\Delta Hmix$ (kJ/mol) | $\Delta H_{mix} = \sum_{i=1,i\neq j}^{n} 4C_i C_j \Delta H_{ij}^{mix}$ | −18.68 |
| $\Omega$ | $\Omega = \frac{T_m \Delta S_{mix}}{|\Delta H_{mix}|}$ | 1.62 |

*4.2. Phase Formation and Strengthening Mechanisms*

The formation of phases in the alloy is influenced by factors such as atomic size, electronegativity, crystal structure, and others. In the studied composition, the atomic size differences of Fe (0.124 nm), Cr (0.125 nm), and Ni (0.124 nm) elements are small and their properties are similar, allowing them to form an infinite solid solution. When the content of Cr is lower than that of Ni, the FCC solid solution based on Fe-Ni is easily formed. Additionally, due to the use of nickel-coated aluminum powder, some Al (0.143 nm) can be dissolved in the matrix, increasing the lattice distortion and resulting in a noticeable solid solution strengthening effect. In comparison Al (0.143 nm), Nb (0.143 nm) and Mo (0.136 nm) atomic radius are larger, may precipitate from the solid solution. In addition, Nb is a strong carbide-forming element, which combines with C that enter the molten pool due to partial melting of the substrate surface during laser cladding. Fine and dispersed carbides (mainly NbC) are synthesized in situ, and also more Mo and Cr are forced to be dissolved in the matrix. The Miedema theoretical model [45] is a theoretical description and calculation of the heat of the formation of binary intermetallic compounds. Some of the calculation results are shown in Table 5. The lower the enthalpy of formation, the easier the intermetallic compounds are to precipitate. Among the compounds formed by Ni and Al, NiAl ($\Delta H_{mix}$ = −48.42 kJ/mol) has a more negative mixing enthalpy than $Ni_3Al$ ($\Delta H_{mix}$ = −33.54 kJ/mol). Therefore, NiAl (B2 phase) precipitation takes precedence in the precipitation sequence. The hard NbC and NiAl, uniformly dispersed in the matrix, contribute to diffusion strengthening. Fe and Nb also tend to form $Fe_2Nb$-type (Laves phase) intermetallic compound due to relatively negative formation enthalpy ($\Delta H_{mix\ Fe2Nb}$ = −20.85 kJ/mol). The FCC matrix phase and Laves phase are simultaneously generated from the liquid phase during the solidification process of the coating, resulting in a distinct fine lamellar eutectic structure. Water cooling treatment prevents immediate grain growth and promotes pseudo-eutectic transformation, leading to a finer and more evenly distributed fine eutectic structure (Figure 4). With the increase in eutectic structure, the number of lamellar boundaries increases, the greater the resistance to prevent dislocation movement, and the more significant the strengthening effect of the lamellar boundary. Therefore, the excellent mechanical properties of the coating can be attributed to the combined effects of solid solution strengthening, diffusion strengthening, lamellar boundary strengthening, and fine-grain strengthening.

**Table 5.** Enthalpy of formation of intermetallic compounds calculated by Miedema theoretical model.

| Compounds | Enthalpy of Formation (kJ/mol) |
|---|---|
| NiAl | −48.42 |
| $Ni_3Al$ | −33.54 |
| $Fe_2Nb$ | −20.85 |

## 5. Conclusions

In this study, the effects of defocusing amount, laser power, scanning speed, and preset powder thickness on the formation quality of AlFeCrMoNi$_{1.8}$Nb$_{1.5}$ EHEA coating prepared by wide-band laser cladding were studied by orthogonal experimental designs. The coating was prepared under the optimum laser cladding process, and the reasons for achieving excellent mechanical properties were explained. The following conclusions are drawn.

The choice of defocusing amount has a significant impact on the quality of coating formation. A smooth surface coating with no holes can be easily obtained by applying negative defocusing. The laser power and scanning speed jointly determine the energy output. Excessive energy output will increase the residual thermal stress of the coating, leading to greater tensile stress upon cooling, which promotes crack formation. The preset powder thickness mainly affects the thickness of the coating. In comprehensive consideration, the influence of the wide-band laser cladding process on the forming quality of the coating from large to small: defocusing amount > laser power > scanning speed > preset powder thickness. The optimum parameters are: defocus amount ($-30$ mm), laser power (4 kW), scanning speed (3 mm/s), and preset powder thickness (1.5 mm). Under this process, the EHEA coating is prepared with good bonding to the substrate and no defects such as holes and cracks.

Water cooling can promote nucleation, refined grains, and the transformation of the pseudo-eutectic zone during the laser cladding, so as to obtain more and finer eutectic structure.

The Ni$_{1.8}$Nb$_{1.5}$ EHEA coating is mainly composed of the FCC matrix phase, Laves phase, B2 phase, and NbC phase. The average microhardness of the coating is 544 HV$_{0.2}$, which is higher than that of the substrate M2 HSS (~220 HV$_{0.2}$). The friction coefficient and wear volume loss of the coating are significantly lower than that of M2 HSS. The wear mechanism of the coating is mainly adhesive wear and slight abrasive wear, which is obviously better than M2 HSS.

The strengthening mechanism of Ni$_{1.8}$Nb$_{1.5}$ EHEA coating in hardness and wear resistance is the synergistic effect of solid solution strengthening, diffusion strengthening, lamellar boundary strengthening, and fine grain strengthening.

**Author Contributions:** Writing—review and editing, F.L., S.Z. and F.Z.; writing—original draft preparation, F.L.; project administration, F.L., S.Z. and F.Z. All authors have read and agreed to the published version of the manuscript.

**Funding:** This research was funded by [the National Natural Science Foundation of China] grant number [No. 51965010], [the Nature Science Foundation of Guizhou Provincial Science and Technology Department] grant number [No. (2020)1Y202], and [Talent Introduced of Guizhou University] grant number [No. (2019)44]. The APC was funded by [Fang Zhou].

**Institutional Review Board Statement:** Not applicable.

**Informed Consent Statement:** Not applicable.

**Data Availability Statement:** Data underlying the results presented in this paper are not publicly available at this time but may be obtained from the authors upon reasonable request.

**Conflicts of Interest:** The authors declare no conflict of interest.

## Abbreviations

| | |
|---|---|
| $\Delta V$ | Wear volume |
| S | Wear trace cross-section area |
| $l$ | The length of the wear track; $l = 4mm$ |
| G | The temperature gradient |
| R | The solidification rate |

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
