# Peer review of "Study on Mechanical Properties of AlFeCrMoNi1.8Nb1.5 Eutectic High-Entropy Alloy Coating Prepared by Wide-Band Laser Cladding"

_coatings, doi:10.3390/coatings13061077_

Round 1

Reviewer 1 Report

Dear authors,

The reviewed work Study On Mechanical Properties of AlFeCrMoNi1.8Nb1.5 Eutectic High-Entropy Alloy Coating Prepared By Wide-band Laser Cladding (coatings-2411365) concerns an important area of science, namely design and manufacture of new materials with special properties, which are undoubtedly HEA alloys. Striving for the practical application of these materials by producing coatings that improve the performance of other materials is undoubtedly the right way. And the use of a laser for this purpose creates such possibilities.

Overall, the article is based on a satisfactory literature review. Serious errors in the research and editorial process were avoided. The only drawback seems to be the too modest style of expression, leading to some ambiguities. I recommend making the following corrections:

19, 79, 226, 277, 380 The microhardness … 544,46 HV02… - There is some confusion in the article related to the recording of measured values, especially microhardness. What is the point of writing to 2 decimal places if the standard deviation of this value is written to 1 decimal place? This is an error. In addition, right next to it there is a record of values without decimal places. Please review and correct this throughout the text.

42, 122, … - …by orthogonal experimental… - What does it mean in practice? The word "orthogonal" is used here as a "smart dodge", which means little, but gives the impression of "smart content". But as I understand it, it is the author who doesn't know something and covers himself up with it.

45 -…scanning speed is 45 mm/s… - Here is a very detailed enumeration, in the next sentence we have only generalities. Please, unify it.

50-52 - What do the ">" signs mean? This is a mental shortcut not explained in the text.

56 - … negative/positive defocus… - What does it mean? Here, where these terms appear, their meaning is not explained. Only further on do I guess this from Fig. 1b. Similarly, in a moment we have: a ring, circular or rectangular distribution. I suggest making a suitable diagram.

100 - The article does not clearly explain the relationship between the notation "AlFeCrMoNi1.8Nb1.5" and "Ni1.8Nb1.5". Is it the same, why are they used interchangeably, etc. Similarly, Table 1 does not explain that the M2 steel is "W6Mo5Cr4V2".

107-116 - In the description of the sample preparation methodology, details regarding the gradation of abrasive papers and polishing materials were omitted or the roughness up to which the process was carried out was not specified.

Table 1 - HEA alloys have a relatively equal atomic fraction of elements, and giving only the weight division is insufficient from this point of view.

127 - Three levels were taken for each factor, regardless of the interaction between the factors - This is not fully understood.

142-148 - The methodology of the wear test does not provide details on the scheme-method of carrying out this test.

Fig. 2 - The photos show that the surface of the substrate was prepared quite accidentally. Maybe it had no effect on the end result, but it is a mistake. For this, it needs proper commentary.

174 - … the coating thickness increases with the decrease of defocusing amount - Not true. Looking at Fig 3, it is clear that one time it goes up and then it goes down. It needs to be better described.

180-182 - …molten pool… - Style. This phrase was repeated three times in one sentence.

Fig. 3 - What does the "T" symbol in the picture mean?

192 - … of insufficient melting occurs… - It has not been explained why, despite the proper penetration of the layer, its defects occur? Perhaps they were not as optimal after all?

207, 223 - … optimal laser cladding process parameters - What are the parameters? Where is it recorded and justified?

218 - … pseudo-eutectic transformation … more eutectic structures… - What do the authors mean by these terms? This needs to be explained and recorded.

226 – bonding zone - What do the authors mean by these terms? This needs to be explained and recorded.

Fig. 5 - The caption to the figure should explain what the acronyms BZ and HEZ mean.

Fig. 6 - In fact, you cannot see anything from these curves, they are either too small or redundant. Please write BZ with a full phrase, just like the rest.

Fig. 8a - This figure is a repetition of Fig. 5 - it is an error and should be corrected or the graph removed from one of these places.

Reviewer 2 Report

The laser cladding process parameters were considered in the manuscript consist of defocusing amount, laser power, scanning speed, and present powder thickness. Some comments given in this format.

1.      Line 137, why should use XRD? Not have another option for microstructure test?

2.      Line 148, the authors use “Contour GT-K three-dimensional optical profiler”, please give further explanation about it.

3.      Line 247, what is the criteria for “strong metallurgical bonding”?

4.      Line 251 in Figure 6, difference of 200 and 20 micro meter in one figure fit differences needs to explained.

5.      What is the novel bought by the authors in the current submission? Its works have been widely discussed in the past. Nothing something really new in the present form. The lack of a novel seems to make the present submission like to replication/modified work. The authors need to detail their novelty in the introduction section. It is a major concern for rejecting this paper.

6.      The work, novelty, and constraints of relevant previous literature must be explained in the introduction section to highlight the article gaps that the present work aims to fill.

7.      Information to the conclusion by formatting it as a paragraph rather than the manner as point-by-point.

8.      The authors encouraged to explain potential further metals study in performing computational simulation. It bring several advantages such as lower cost and faster results compared to experimental test as conducted in the present study. Also, for this purpose, rever the relevant reference as follows: Tresca Stress Study of CoCrMo-on-CoCrMo Bearings Based on Body Mass Index Using 2D Computational Model. Jurnal Tribologi 2022, 33, 31–8. https://jurnaltribologi.mytribos.org/v33/JT-33-31-38.pdf

-

Reviewer 3 Report

My comments are attached

I recommend that the authors thoroughly review the English language usage in their work.

Reviewer 4 Report

Numbers means line in the article.

107: delete space after "x" sign and 10.

116: delete space after ":" sign and 3.

132: put space after sign "," and 0.

143: small leter of time unit: "s" instead of "S".

173: delete dot after "3". Remark for the rest of similar situations (e.g. line 222, 274).

220: It will be better if yellow font on the pictures has other color, more visible.

294: in values put proper sign, not letter "x".

The article has good quality and clearly form of presentation. Researches are described properly. Figures has good quality with good descriptions.
